# Immunotherapy in Head and Neck Cancer When, How, and Why?

**DOI:** 10.3390/biomedicines10092151

**Published:** 2022-09-01

**Authors:** Daniela Pereira, Diana Martins, Fernando Mendes

**Affiliations:** 1Politécnico de Coimbra, ESTeSC, UCPCBL, Rua 5 de Outubro–SM Bispo, Apartado 7006, 3046-854 Coimbra, Portugal; 2Laboratório de Investigação em Ciências Aplicadas à Saúde (LabinSaúde), Politécnico de Coimbra, ESTESC, Rua 5 de Outubro–SM Bispo, Apartado 7006, 3046-854 Coimbra, Portugal; 3Coimbra Institute for Clinical and Biomedical Research (iCBR) Area of Environment Genetics and Oncobiology (CIMAGO), Biophysics Institute of Faculty of Medicine, University of Coimbra, 3000-548 Coimbra, Portugal; 4Center for Innovative Biomedicine and Biotechnology (CIBB), University of Coimbra, 3004-504 Coimbra, Portugal; 5Clinical Academic Center of Coimbra (CACC), 3004-561 Coimbra, Portugal; 6European Association for Professions in Biomedical Sciences, B-1000 Brussels, Belgium

**Keywords:** head and neck neoplasms, therapeutics, immune checkpoint inhibitors, cancer vaccines, oncolytic viruses

## Abstract

Head and neck cancer (HNC) is one of the most common cancers worldwide. Alcohol and tobacco consumption, besides viral infections, are the main risk factors associated with this cancer. When diagnosed in advanced stages, HNC patients present a higher probability of recurrence or metastasising. The complexity of therapeutic options and post-treatment surveillance is associated with poor prognosis and reduced overall survival (OS). This review aims to explore immunotherapy (immune checkpoint inhibitors (ICI), therapeutic vaccines, and oncolytic viruses) in HNC patients’ treatment, and to explore when, how, and why patients can benefit from it. The monotherapy with ICI or in combination with chemotherapy (QT) shows the most promising results. Compared to standard therapy, ICI are able to increase OS and patients’ quality of life. QT in combination with ICI demonstrates significant response rates and considerable long-term clinical benefits. However, the toxicity associated with this approach is still a hurdle to overcome. In parallel, the therapeutic vaccines directed to the Human Papilloma Virus are also efficient in increasing the antitumour response, inducing cellular and humoral immunity. Although these results demonstrate clinical benefits compared to standard therapy, it is also important to unravel the resistance mechanisms in order to predict the clinical benefit of immunotherapy.

## 1. Introduction

Head and neck cancer (HNC) is the 6th most common neoplasm worldwide, with a higher incidence in developing countries [1,2]. In total, 90% of the HNC cases are head and neck squamous cell carcinomas (HNSCC) [3,4]. HNSCC derives from the squamous epithelium of the upper air-digestive tract, affecting different anatomical areas, such as the oral cavity, oropharynx, larynx, and hypopharynx [4,5,6].

The consumption of tobacco, alcohol, and chronic infection by the Human Papilloma Virus (HPV) are the main risk factors associated with the development of HNC [1,7,8,9]. Approximately 75% of HNSCC cases are associated with tobacco and alcohol consumption, inducing mutations in the tumour protein p53 gene (*TP53*), interfering with deoxyribonucleic acid (DNA) synthesis and DNA repair mechanisms [1]. Thus, proliferation, angiogenesis, and migration of tumour cells are potentiated [1]. More than 80% of HNSCC patients have mutations in *TP53*, which lead to a reduction in overall survival (OS), resistance to therapy, and an increased recurrence rate [8]. However, recently, the incidence of HNSCC associated with HPV infection has been increasing, especially at young ages [10]. Within the different genotypes, the World Health Organization (WHO) considers HPV-16 the most frequent type associated with oropharyngeal HNSCC, exerting its carcinogenic effect through two viral oncoproteins—E6 and E7 [1,10,11]. These oncoproteins assume the ability to degrade two tumour suppressors, P53 and retinoblastoma protein (pRb), promoting uncontrolled proliferation, genomic instability, and cell cycle dysregulation [1,8,9,11]. Epstein–Barr (EBV) infection may also trigger cancer in the nasopharynx due to the encoding of latent membrane proteins, such as latent membrane proteins 1 and 2 (LMP1 and LMP2), contributing to the progression of HNSCC [12,13].

When diagnosed, more than half of the patients are already in advanced stages of neoplasia (III or IV), and 10% of patients have already developed metastases [9,14]. Within the different HNSCC variants, HNSCC HPV-negative is the most frequent, with mutations at *TP53*, cyclin-dependent kinase inhibitor 2A (*CDKN2A*)*,* phosphatidylinositol-4,5-bisphospate 3-kinase catalytic subunit alpha (*PIK3CA*)*,* FAT atypical cadherin 1 (*FAT1*), and NOTCH receptors (*NOTCH*) due to tobacco and alcohol consumption [15].

The most common therapeutic approaches include surgical resection of the primary tumour and ganglia drainage, followed by adjuvant radiation, with or without platinum-based chemotherapy (QT). Concurrent chemoradiation and radiation therapy are used for early and advanced cancers [3,7,8,14,16]. Despite the advances in therapy, patients with advanced disease demonstrate a 5-year OS of 10–50% and patients with recurrence or metastases have a worse prognosis [14,17,18]. For that reason, it is necessary to develop different therapeutic approaches that increase the survival of HNSCC.

With the development of monoclonal antibodies (mAb), the Food and Drug Administration (FDA) approved in 2006 the first mAb directed at the epidermal growth factor receptor (EGFR) in patients with HNSCC—cetuximab [19,20]. For that reason, platinum-based QT with cetuximab has become the standard therapy for patients with recurrent and/or metastatic local disease, resulting in increased patient survival and a reduced risk of death [14,20]. However, response rates of HNSCC patients to targeted therapy remain low, suggesting that tumour heterogeneity on a molecular and immunological level may be associated with these results [21].

In recent years, continuous research on carcinogenesis has established an association between the tumour and the host immune system, where modifications in immunological surveillance and the tumour microenvironment (TME) promote the evasion of tumour cells into the immune system (Figure 1) [7,20]. Immunotherapy was capable of stimulating the host immune system, promoting the lysis of tumour cells and consequent tumour regression, minimising damage to normal cells [9,20,22]. Immune checkpoint inhibitors (ICI), vaccines, and oncolytic viruses for the treatment of HNSCC patients have gained particular prominence [3].

To stratify and direct patients to the best immunotherapeutic approaches, it is important to understand the genetic abnormalities and mechanisms of tumour evasion [23]. To evade the immune system, these tumours negatively regulate the machinery via the class I human leukocyte antigen (HLA) system through genetic and epigenetic alterations, as well as inactivating the antigen processing machinery [21]. In addition, tumours induce T-cell apoptosis, recruit immunosuppressive cells such as regulatory T (Tregs), myeloid-derived suppressor cells (MDSC), and M2 macrophages, and upregulate inhibitory checkpoint molecules (Figure 2) [6,21,23,24,25,26]. The presence of certain enzymes, such as indoleamine 2,3 dioxygenase 1, has the ability to suppress essential amino acids for T cell proliferation, such as tryptophan [27]. It is also important to consider that tumour cells are still under a dynamic process, designated immunoediting, where the least immunogenic cells are positively selected, contributing to tumour progression [24,28].

The high mutational rate with neoantigen formation and a highly inflamed TME also suggests a possible positive response to immunotherapy [6,15,29]. However, it is necessary to consider HPV-associated tumours because a chronic viral infection can originate from a single and non-self-antigenic target [10,15]. Furthermore, it was observed that a greater rate of immune infiltration and programmed death-ligand 1 (PD-L1) expression in the tumour tissue [10,15]. Based on the above, biomarkers such as PD-L1, HPV status, and TME may help in the selection of therapy and evaluation of the progression of premalignant lesions and the prognosis of neoplasia [7,16].

### 1.1. Immunotherapy

#### 1.1.1. Immune Checkpoint Inhibitors

Immune checkpoints are molecules (co-stimulators and inhibitors) that are fundamental for the regulation of immune homeostasis, mediating the immune response of T cells [30]. These molecules prevent the hyperreactivity of the immune system and the development of possible autoimmune diseases [30]. However, tumour cells assume particular characteristics that allow them to evade the immune system by activating co-inhibitory pathways, thereby inducing immune tolerance and tumour progression [25,29].

Within these pathways, molecules such as programmed death-1 (PD-1) and cytotoxic T lymphocyte antigen 4 (CTLA-4) increase relevance by operating in different stages of T-cell activation and with different mechanism inhibition [30,31,32,33,34].

##### Programmed Death-1 and Programmed Death-Ligand 1

PD-1, also known as CD279, is a transmembrane protein encoded by the programmed cell death protein 1 (*PDCD1*) gene on chromosome 2 [22,35]. This molecule is expressed in activated CD4^+^ and CD8^+^ T-cells, natural killer (NK) cells, B cells, tumour-associated macrophages (TAM), dendritic cells, and monocytes, interacting with the ligand present in antigen-presenting cells (APC) and tumour cells (TC) [22,25,30]. PD-1 features two known ligands, namely, programmed death-ligand 1 and 2 (PD-L1 and PD-L2) [36]. Unlike PD-L1, PD-L2 is more restricted to APC and is not widely expressed in the tumour parenchyma of HNSCC patients, suggesting that it is not a target of interest for current immunotherapy [36].

PD-L1, also known as CD279, is a type 1 transmembrane glycoprotein encoded by the *CD274* gene, widely expressed in tumour cells, and weakly expressed in some activated T and B cells and dendritic cells [22,35,37].

About 50–60% of HNSCC patients express PD-L1, which is induced by the inflammatory environment in TME—the presence of interferon-gamma (IFN-*γ*), tumour necrosis factor-alpha (TNF-*α*), granulocyte-macrophage colony-stimulating factor (GM-CSF) and interleukin (IL) 4 [15,18,23,36,38]. Thus, the PD-1/PD-L1 interaction inhibits the proliferation, cytokine production, and survival of T cells—phenotypic typical of exhaustion, conditioning the antitumour response mediated by CD8^+^ T cells [17,18,29,30]. In addition, the phagocytosis capacity of the TAM present in the TME is also compromised, thus increasing tumour immunity [30].

The use of anti-PD-1/PD-L1 mAb has been shown to restore T-cell-mediated antitumour immunity—FDA-approved pembrolizumab and nivolumab in 2016 for patients with recurrent, metastatic, and unresectable HNSCC [8,19,22,36,39]. durvalumab and atezolizumab are performing clinical trials aiming at the same objective as the previous [3,22].

##### Cytotoxic T Lymphocyte Antigen 4

CTLA-4, also known as CD152, is a protein encoded by exon-4 of the CTLA4 gene [3]. This molecule is essentially expressed in Treg cells and activated CD4^+^ and CD8^+^ T cells, being involved in stopping the co-stimulatory signal of T cells [30,35]. In naïve T cells at rest, CTLA-4 is found in the cytoplasm, which, after the T-cell receptor (TCR) stimulus, is translocated to the cell membrane and is now expressed to the cell surface [6,38].

This protein is homologous to CD28 (co-stimulating molecule), also expressed in T cells that share the same binding—CD80 and CD86 (molecules B7-1 and B7-2) expressed in APC [3,25,30,40]. During the immune response, dendritic cells present antigens to CD4^+^ T-cells that become T helper 1 (Th1) cells, producing IFN-*γ*, TNF-*α*, and IL-2 that promote CD8^+^ T-cell proliferation [30,41]. However, when APC interacts with CTLA-4, the production of IFN-*γ*, TNF-*α*, and IL-2 is reduced, resulting in the suppression of T-cell proliferation [30,41], suggesting that CTLA-4 has greater avidity and affinity for the ligands compared to CD28 [25,29]. Tumour cells also secrete transforming growth factor-beta (TGF-β) capable of inducing the expression of CTLA-4 on the surface of T cells, leading to their exhaustion [25]. The use of anti-CTLA-4 mAb, such as ipilimumab and tremelimumab, may be a promising strategy in the treatment of several cancers, including HNSCC [19].

##### Natural Killer Cell Receptor

In addition to the PD-1/PD-L1 and CTLA-4 inhibitory pathways, others have been prominent in immunotherapy, such as receptors with inhibitory motives based on intracytoplasmic tyrosine (ITIM) [42]. These receptors can recruit phosphatases (Scr homology region 2-containing protein tyrosine phosphatase (SHP)-1/2 or Src homology 2 domain-containing inositol 5′-phosphatase (SHIP)) that transmit the inhibiting signal to effector immune cells (ICs) [42].

NK group 2 member A (NKG2A) is an ITIM transporter receptor expressed as a heterodimer with CD94 in NK cells and a subgroup of CD8^+^ T [42,43]. Its immunosuppressive effect is exerted from the interaction between NKG2A/CD94 and the non-classical HLA class I histocompatibility antigen, alpha chain E-(HLA-E) molecule overexpressed in these tumours, inhibiting the effector functions of CD8^+^ T and NK cells after phosphatase SHP-1 recruitment [42].

Thus, mAb monalizumab prevents the binding of NKG2A/CD94 to HLA-E, usually induced by antitumour activity and IFN-*γ*, resulting in the increased cytotoxic activity of NK and T CD8^+^ cells [43].

#### 1.1.2. Vaccines

Therapeutic vaccines aim to stimulate the immune system to develop a specific immune response from the administration of known tumour neoantigens [6,44]. Tumour-associated antigens, weakly expressed in normal tissues, are still a barrier to the development of therapies as they may induce autoimmune toxicity in normal tissues [45]. Therefore, tumour-specific antigens are the most attractive for immunotherapy [45].

In the case of HNSCC, viral oncoproteins E6 and E7 are the most promising biological targets in the development of immunotherapeutic vaccines since they are constitutively expressed in tumour cells and can be recognised by the immune system as non-self-promoting the antitumour response [3,10].

In this therapy stand out peptide vaccines (ISA-101 and P16_37-63), viral vector-based vaccines (recombinant serotype 5 adenovirus (rAd5)-EBV-LMP2), and nucleic acid vaccines (MEDI0457) [3,10,46].

##### Peptide Vaccines

Peptide vaccines are composed of peptides captured and processed by APC and subsequently presented by HLA molecules of class I and II to naïve T cells in the lymph nodes, resulting in stimulation of CD4^+^ and CD8^+^ T cells for an antitumour response [46,47,48].

The ISA-101, directed to HPV-16, consists of long HPV-E6 and HPV-E7 peptides aiming at T-cell stimulation inducing the antitumour response [10,45]. In turn, the P16_37-63, consisting of a peptide of 27 amino acids in length, targets the viral antigen p16^INK4a^ [47]. This antigen is widely expressed in infected tissues, having been induced by the activation of the histone lysine demethylase 6B, mediated by oncoprotein E7, and epigenetic reprogramming of the *CDKN2A* gene [47].

Although the stability, safety, and easy productivity of these vaccines are of interest to the clinic, it should be noted that they are restricted to the major histocompatibility complex (MHC). Due to the heterogeneity of HLA molecules among individuals, it becomes difficult to develop comprehensive vaccines for the entire population [44].

##### Vaccines Based on Viral Vectors

The viral vectors used in these vaccines can invade host cells, such as dendritic cells, replicating themselves and inducing the encoding of tumour antigens presented in MHC class I and II for T cells [44].

rAd5-EBV-LMP2 is a vaccine based on a viral vector, consisting of a rAd5 carrying EBV-LMP2 [12]. The administration of this therapeutic aims to increase the proliferation of antigen-specific T cells and, consequently, the immune cytotoxicity against tumour cells of patients with Epstein–Barr-associated nasopharynx cancer (NPC) [12].

##### Nucleic Acid-Based Vaccines

Other types are vaccines that consist of DNA or messenger ribonucleic acid plasmids capable of replicating and encoding the proteins of interest for immunotherapy in host cells [4]. This way, the transfected gene will encode the tumour antigen processed and presented in the MHC class I and II for naïve T cells [4].

MEDI0457 is a DNA vaccine containing synthetic plasmids targeting the E6 and E7 antigens of HPV-16 and HPV-18 genotypes, coupled with IL-12 coding genes to increase the immunogenicity of the vaccine [10,49]. IL-12 is a cytokine promoting the maturation and functionality of Th1 cells, enhancing the innate and acquired immune response [27,49].

#### 1.1.3. Oncolytic Viruses

Oncolytic viruses are capable of selectively infecting tumour cells with specific receptors, causing their lysis without harming normal host cells [3,6,50]. Its antitumour effect can be exercised directly or indirectly by the release of pathogen-associated molecular patterns, damage-associated molecular patterns, and tumour-associated antigens that stimulate the specific immune response, contributing to the reduction of large tumours [3,6].

Herpes simplex viruses (HSV) contain a mostly non-coding genome, allowing the manipulation and addition of transgenes, with subsequent endocytosis in tumour cells [3]. Within these, Talimogene laherparepvec (T-VEC), approved for the treatment of patients with melanoma, and HF10, still in an in vitro assays, are highlighted [3].

T-VEC is a doubly mutated HSV-1 with dissections in the *γ34.5* and *α47* genes, and with insertion of the gene encoding the GM-CSF in the loci *γ34.5* [50,51]. This virus is capable of infecting and replicating in tumour cells by cell receptors such as nectins, glycoproteins, and herpesvirus entry mediators [50,52]. In addition, its replication is also associated with the interruption of other oncogenic signalling pathways, such as protein kinase R and IFN type I pathways [52]. Antitumour immunity is believed to be enhanced by GM-CSF, which is capable of recruiting dendritic cells to inflamed sites, promoting APC functionality, and further stimulating T-cell response [52]. HF10 is a naturally mutated virus in the *UL56* gene capable of replicating and killing tumour cells, as well as suppressing tumour growth in patients with HNSCC [3].

Different clinical trials have demonstrated the possible beneficial effect of immunotherapy in different treatment scenarios as well as when combined with other therapies.

It is clear that recognising the different subtypes of HNC will become essential to selecting the best therapeutic strategy. To the present date, no discrepancies have been found between the treatment administered to HPV-positive and HPV-negative patients [53,54]. However, it is important to bear in mind that HPV-positive patients tend to have better responses to different therapeutic approaches, suggesting the therapy can be optimised in order to reduce the toxicity commonly associated with chemotherapy [54].

Based on the evolution of immunotherapy, the administration of pembrolizumab and nivolumab in recurrent/metastatic (R/M) HNSCC patients has been approved [14]. In 2019, the combination of pembrolizumab with platinum and fluorouracil was also approved for patients with metastatic, unresectable, or recurrent tumours, regardless of PD-L1 expression, as well as the administration of pembrolizumab alone in patients with a combined positive score (CPS) ≥ 1 regarding PD-L1 expression [14].

This systematic review aims to explore different immunotherapies, such as ICI, therapeutic vaccines, and oncolytic viruses, in the treatment of patients with HNC. Identifying the immunological and molecular patterns of these tumours will clarify when, how, and why immunotherapy should be used in HNC treatment.

## 2. Materials and Methods

The authors used the Preferred Reporting Items for Systematic Reviews and Meta-Analyses (PRISMA) statement to perform this systematic review to better understand the role of immunotherapy in head and neck cancer [55].

From a selection of the eligible literature, it will become possible to understand when, how and why different immunotherapies should be administered to these patients.

To achieve this goal, a literature search was carried out using the PubMed database. In this database, the keywords used for the research were “head and neck neoplasms”, “therapeutics”, “immune checkpoint inhibitors”, “cancer vaccines”, and “oncolytic viruses”. Table 1 presents the research strategies used, based on the combination of keywords (Medical Subject Headings terms), to obtain the literature included in this review.

In this research, all selected articles were properly identified, analyzed, and selected based on inclusion and exclusion criteria. The definition of these criteria ensures that all articles selected for study present accurate and relevant information about the addressed theme. At PubMed, at the level of text availability, free full texts were selected, making clinical trials, meta-analyses, randomised clinical trials, reviews, and systematic reviews written in English and published between 2016 and 2022 eligible. In parallel, the exclusion criteria refer to all literature that does not meet the above-mentioned criteria. The investigation strategies applied in this systematic review are detailed in Figure 3, and all the sources are accordingly referenced. This systematic review is registered at the International Platform of Registered Systematic Review and Meta-analysis Protocols (INPLASY) with the digital object identifier (DOI) number 10.37766/inplasy2022.8.0016.

## 3. Results

At the time of diagnosis, most patients are in advanced stages with limited therapeutic options and short overall survival rates. In this sense, it is important to address new therapeutic strategies, such as ICI, vaccines, and oncolytic viruses, to promote better survival and response rates in these patients. This review included 21 articles (clinical trials, randomised clinical trials, and reviews) in the field of immunotherapy, published between 2016 and 2022, in patients with HNC.

The KEYNOTE-012 phase Ib is a multicenter study with two study cohorts (initial versus expansion) with the aim of evaluating the activity of pembrolizumab at different doses in patients with R/M HNSCC (Table 2). Within the 192 enrolled patients, 77% (n = 147) were HPV-negative and 23% (n = 45) were HPV-positive. Regarding the expression of PD-L1, 81% presented a CPS ≥ 1, while in 19% the score was less than one. In the treated population, 4% of patients had a complete response (CR), while 14% had a partial response (PR), resulting in an objective response rate (ORR) of 18%. It is estimated that about 85% of responses lasted ≥ 6 months and 71% of responses lasted ≥ 12 months, thus demonstrating the long durability of this mAb. The average rates of progression-free survival (PFS) and OS were 2.1 months and 8 months, respectively. In the treated population, 13% had grade 3/4 adverse events (AE), with elevations in alanine aminotransferase (ALT) and aspartate aminotransferase (AST) being the most frequent. Stratifying patients based on HPV status, the ORR rate was higher in the HPV-positive (24%) subgroup compared to the HPV-negative group (16%). The same was also observed in patients with CPS ≥ 1 (21%), compared to patients with CPS < 1 (6%). The average PFS and OS were 2.1 months and 10 months for the group with CPS ≥ 1. For the other subgroup, the same rates were 2.0 months and 5 months [56].

The KEYNOTE-055 is a single-arm, phase II study that evaluated the activity of an anti-PD-1 in patients with R/M HNSCC refractory to platinum-based therapy and cetuximab (Table 2). Of the 171 patients enrolled, 140 were known to have a CPS of between ≥ 1 and 48 had a CPS ≥ 50. Considering HPV, 131 patients were negative and 37 were positive. After evaluation of imaging, ORR was 16% with a CR and 27 PR, with a mean duration of response (DOR) of 8 months. The median PFS and OS were 2.1 months and 8 months. In the treated population, 64% of patients had the following AE: fatigue, hypothyroidism, nausea, and increased AST being the most common. Of these, 15% experienced a grade ≥ 3 and seven patients discontinued treatment. In the HPV subgroups, the ORR and PFS were similar; however, HPV-positive patients demonstrated a higher 6-month OS (72%), compared with 55% in the HPV-negative subgroup. Based on PD-L1 expression, patients with higher expressions tend to demonstrate a higher ORR rate (18% (CPS ≥ 1) and 27% (CPS ≥ 50) compared to patients with expressions lower than one and fifty [57].

Based on clinical trials KEYNOTE-012 and KEYNOTE-055, Cohen et al. [58], initiated a phase III study, KEYNOTE-040, comparing the efficacy and safety of pembrolizumab with standard therapy in patients with R/M HNSCC who progressed after platinum therapy (Table 2). Based on PD-L1 expression, 79% of patients treated with pembrolizumab and 77% of patients treated with standard therapy had CPS ≥ 1. An ORR of 14.6% of patients treated with mAb was observed, within which 1.6% had a CR and 13% had PR. Patients undergoing standard therapy had an ORR of 10.1%, so 0.4% had a CR and 9.7% had a PR. The mean response time was 4.5 months and 2.2 months for patients treated with pembrolizumab and standard therapy, respectively. The median PFS was 2.1 months in patients treated with mAb and 2.3 months in patients undergoing standard therapy. For the same groups, the median OS was 8.4 months and 6.9 months. The median OS considering CPS ≥ 1 and TPS ≥ 50% was 8.7 months and 11.6 months in patients treated with pembrolizumab and 7.1 months and 6.6 months in patients treated with standard therapy. Hypothyroidism, fatigue, diarrhoea, and skin irritation were the most common AEs observed. Grade 3-5 AE occurred in about 13% and 36% of patients leading to the discontinuation of 6% and 5% of patients treated with pembrolizumab and standard therapy, respectively [58].

The CheckMate 358 clinical trial evaluated nivolumab activity in the neoadjuvant setting in 52 patients with resectable HNSCC (Table 2). In this study, 26 patients were HPV-positive and the remaining were HPV-negative. Based on the expression of PD-L1, most patients had a CPS ≥ 1 and a TC > 1%. Of the 49 evaluable for radiographic response, 56% of HPV-positive patients and 12% of HPV-negative patients demonstrated tumour reduction, and reductions greater than 30%were only observed in 12% and 8.3%, respectively. Among the 34 tumours pathologically evaluated by central revision, one patient achieved a complete pathological response (MPR, ≤ 10% without residual viable tumour (RVT)), and three achieved a partial pathological response (pPR, 10% < RVT ≤ 50%) in the HPV-positive cohort, resulting in a rate of 23.5%. Only one patient in the HPV-negative cohort reached a pPR. After 24 months post-op, the recurrence-free survival (RFS) rates were 88.2% and 52.2% for HPV-positive and HPV-negative cohorts, respectively. Thirty-six months after the start of treatment, the OS were 100% and 63.5% for HPV-positive and HPV-negative cohorts, respectively. In cohort HPV-negative, the median OS was 49.8 months; however, the same rate was not achieved in the other subgroup. After the last dose of nivolumab, 73.1% and 53.8% of patients from the HPV-positive and HPV-negative cohorts respectively presented an AE of any degree, with fatigue being the most common. For the same subgroups, 19.2% and 11.5% of patients had grade 3/4 AE. Despite this, no patient discontinued treatment [59].

CheckMate 141 is a phase III clinical trial developed to assess nivolumab (anti-PD-1) activity in 361 R/M HNSCC patients who progressed after platinum-based QT over 6 months (Table 2). Considering ORR rate, the group administered with nivolumab demonstrated a higher rate (13.3%), with 6 CR and 26 PR, compared with the group administered with standard therapy (5.8% with only 1 CR and 6 PR), with a mean response time of 2.1 and 2.0 months, respectively. Considering the median PFS, the results did not show statistically significant differences between the two groups; however, 6-months PFS esteemed were 19.7% and 9.9%, which revealed a significant difference. The median OS was 7.5 months and 5.1 months for patients treated with nivolumab and standard therapy, respectively. The AE were similar between the two groups; however, the standard therapy group presented a grade of 3/4 AE higher than the nivolumab group (35.1% versus 13.1%). The most common AEs were fatigue, nausea, rash, and decreased appetite. Considering both therapies, patients undergoing immunotherapy reported a higher quality of life, some with minor improvements, compared to the remaining patients in the study. Relatively HPV status, median OS in patients treated with nivolumab was higher in HPV-infected patients (9.1 months) than in patients without HPV (7.5 months). In patients treated with standard therapy, median OS was similar in HPV-positive versus HPV-negative patients (4.4 months versus 5.8 months). Based on TC expression of PD-L1, patients treated with immunotherapy show a higher ORR (15% (TC > 1%) and 9% (TC < 1%)) rate compared with patients undergoing standard therapy [60].

In the neoadjuvant scenario, a phase II clinical trial evaluated nivolumab activity with an anti-CTLA-4 agent compared to nivolumab monotherapy in 29 patients with squamous cell carcinoma in the oral cavity (OC-SCC) (Table 2). Regarding tumour response, both treatment arms (nivolumab versus nivolumab with ipilimumab) demonstrated evidence of it as follows: volumetric response of 50% versus 53%, pathological downstaging of 69% versus 53%, and response evaluation criteria in solid tumours (RECIST) response of 13% versus 38%, respectively. The pathological response of the tumour (PTR) was based on a quantitative classification scheme, being that 38% and 40% of patients treated with nivolumab and nivolumab more ipilimumab had a PTR1 (≥ 10% and <50% response), while 15% and 33% of patients treated with the same strategies had a PTR2 (≥50% response). An average follow-up of 14.2 months, 1-year PFS, and OS were 85% and 89%, respectively were observed. In what concerns safety, 21 patients (72%) revealed AE possibly associated with treatment under study. Moreover, 2 patients treated with nivolumab and 5 patients treated with both mAbs had grade 3/4 AE [61].

In 2021, McBride et al. [62] published the results of a randomised phase II study evaluating immunotherapy activity (nivolumab) combined with stereotactic body radiation therapy (SBRT) compared to nivolumab monotherapy in 62 patients with metastatic HNSCC (Table 2). Statistically, there were no significant differences between the two treatment arms, so monotherapy demonstrated a slightly higher ORR (34.5%) compared to the arm treated with nivolumab and SBRT (29%). Median DOR was observed only in the arm administered with mAb and SBRT, which was 9.4 months. The median PFS in arms treated with nivolumab or nivolumab and SBRT was 1.9 months and 2.6 months. The median OS was 14.2 months and 13.9 months, respectively. About 13.3% of patients in the monotherapy arm and 9.7% in the other arm had grades 3-5 AE. In a more intrinsic analysis of the 60 patients with ORR data, HPV-positive patients had a higher ORR (41.9%) compared to HPV-negative patients (20.7%), in addition to also presenting a higher 1-year PFS (64.4%) compared to the second subgroup (40.5%). For the 56 patients with positive expressions of PD-L1 (TC ≥ 1%), the ORR was higher (50%) compared to PD-L1-negative (TC < 1%) patients (23.5%). For these patients, 1-year PFS was 63% and 47%, respectively [62].

To evaluate the efficacy and safety of an anti-PD-1 agent (Sintilimab) combined with induction chemotherapy (IC), Li et al. [63] developed a study involving 163 patients with locally advanced and untreated HNSCC (Table 2). After evaluation of the expression of PD-L1 and P16, it is known that 28.6% (n = 16) of patients with oropharynx cancer (OPC) are HPV-positive. In addition, of the 65 patients submitted to immunotherapy, 76.9% had a CPS ≥ 1 and 40% had a CPS ≥ 20. Based on the results presented, the ORR in the arm administered with IC was 68.4% with 10 CR and 57 PR. In the arm treated with Sintilimab and IC, ORR was significantly higher (84.6%) with 15 CR and 40 PR. For the same arms, the 2-year PFS was also a statistically significant difference—27% (IC) versus 44% (IC and Sintilimab). The median OS for both treatment arms was similar; however, the 2-year OS in the subgroup submitted to immunotherapy and IC was higher (70%) compared to the group submitted only to IC (61%). In both treatment arms, all patients manifested grade 1/2 AE. The most common AEs were anorexia, nausea, fatigue, and constipation. Grade 3/4 AE was observed in 15.3% and 18.5% of patients in the arms IC and Sintilimab combined with IC [63].

Docetaxel (DTX) is an agent used in the therapy of patients with R/M HNSCC, and preclinical models have revealed its ability to increase immune response. Based on this evidence, Fuereder et al. [64] developed a single-arm, phase I/II study, assessing the safety and efficacy of DTX combined with pembrolizumab in 22 patients with platinum-resistant R/M HNSCC (Table 2). All patients demonstrated positive PD-L1 expression for a CPS ≥ 1, while 38% specifically had a CPS ≥ 20. The results presented revealed a 3-month ORR of 22.7%, within which one patient had a CR and four had PR. The median PFS and OS were 5.8 months and 21.3 months, respectively. Regarding AE, all patients had myelosuppression, while 13.6% had a grade 3 AE (febrile neutropenia). In addition, one patient had grade 5 AE—immune thrombocytopenia [64].

HAWK is a single-arm, phase II study that evaluated the activity of durvalumab monotherapy in 112 patients with platinum-refractory R/M HNSCC with elevated PD-L1 tumour expression (≥25%) (Table 2). Of the 99 patients evaluated for HPV status, 34 were positive and 65 were negative. The ORR was 16.2% with 1 CR and 17 PR. Approximately 40.5% of the patients presented a reduction in tumour size upon imaging evaluation. The median PFS and OS were 2.1 months and 7.1 months, respectively. It is known that 57.1% (n = 64) of patients had at least one AE of any degree, and 9 patients had a grade of 3/4 AE. The most common AEs were nausea, fatigue, hypothyroidism, and asthenia. Stratification of patients based on HPV status. The ORR for HPV-positive patients was 29.4% and 10.8% among the 65 negative patients. PFS and OS were 3.6 months and 10.2 months for HPV-positive patients, and 1.8 months and 5.0 months for HPV-negative patients [65].

In 2018, Colevas et al. [66] developed a phase I clinical trial evaluating the activity of atezolizumab in 32 patients with advanced-stage HNC (Table 2). This study also considered the expression of PD-L1 in infiltrating ICs (IC_s0/1_ < 5%, and IC_s2/3_ ≥ 5%) and HPV status, however, were not associated with response to mAb. Of the 28 patients evaluated for HPV status, 12 were positive, 13 were negative, and 3 had an unknown status. In the entire population treated, the ORR was 22%, which was all PR, with a median DOR of 7.4 months. The median PFS and OS were 2.6 and 6.0 months. Within subgroups IC_s0/1_ (n = 7) and IC_s2/3_ (n = 25), ORR was 14% and 24%, with a median DOR of 7.4 months and 26.2 months, respectively. The median PFS and OS in both were similar to the general population in this study. Based on HPV status, ORR was 17% and 15%, with all PR, for HPV-negative and HPV-positive patients, respectively. Administration of the mAb was well tolerated, with 66% of patients having AE of any degree, with fatigue and rash being the most common. Only 13% of patients had grade 3/4 AE. The questionnaires evaluating patients’ quality of life demonstrated positive results concerning therapy [66].

The Phase I JAVELIN Solid Tumour trial evaluated avelumab activity in 153 patients with R/M HNSCC refractory to platinum therapy and never treated with ICI (Table 2). With a mean follow-up of 27.9 months at the time of data cutting, ORR was 9.2% evaluated by a blinded independent review committee (ICR) (2 patients had CR and 12 PR) and 13.1% by the investigator (5 patients had CR and 15 PR). The median DOR was not achieved by the evaluation of IRC; however, the evaluation allowed follow-up beyond cutting data, revealing a median DOR of 30.4 months. The median PFS, evaluated by IRC and the investigator, was 1.4 months and 1.8 months, respectively. On the other hand, the median OS was 8.0 months. The results revealed that 83 patients had AE of any degree, and 10 had grades ≥3. Fatigue, fever, pruritus, and chills were the most common AEs. The grade 3 immune-mediated AE (imAE) was also observed in three patients, leading to the discontinuation of treatment in one patient. Stratifying patients based on HPV-positive and HPV-negative patients, ORR was 15.4% (ICR) versus 17.9% (investigator) and 5.1% (IRC) versus 11.1% (investigator). For the same groups, the median PFS was 2.7 versus 3.3 months and 1.4 versus 1.4 months, and the median OS was 11.8 months and 7.4 months. Patients with positive expression of PD-L1 had a higher ORR (10.3% (IRC) versus 15% (investigator), compared to patients with negative expression (3.3% (ICR) versus 6.7% (investigator)). For these patients, median OS was 7.9 months versus 8.9 months [67].

CONDOR studied the activity of dual immunotherapy with anti-PD-L1 (durvalumab) and anti-CTLA-4 (tremelimumab), compared to monotherapy with these same mAbs. (Table 2). The 267 enrolled patients had R/M HNSCC with the progression of the neoplasm after a platinum regimen and low/negative tumour expression of PD-L1 (<25%). Most patients (52.9%) had tumour expression of PD-L1 < 1%, while 30% had 1% ≤ PD-L1 < 10% and 17.1% had 10% ≤ PD-L1 < 25%. Regardless of the primary site of the tumour, all patients were evaluated by HPV status, identifying 75 positive cases. Regarding the ORR rate in these patients, durvalumab monotherapy showed a higher ORR (9.2%) compared to dual immunotherapy (7.8%) and tremelimumab monotherapy (1.6%), all of which were PR. The median DOR was 9.4 months in the arm treated with dual therapy, and it was not achieved in the other two arms. The median PFS in the arm treated with durvalumab and tremelimumab was 2.0 months and 1.9 months in each of the monotherapy arms. The results of the trial also showed that patients administered with both mAbs had a higher median OS (7.6 months) compared to durvalumab monotherapy (6.0 months) and tremelimumab (5.5 months). The AE were similar in the three arms, so diarrhoea, fatigue, and asthenia were the most common. Grade 3/4 AE occurred in 15.8%, 12.3%, and 16.9% in the dual therapy, durvalumab, and tremelimumab arms, respectively. In addition, imAE was reported in 19.5% and 7.7% of patients undergoing combination therapy or durvalumab, respectively. As a result, twelve patients discontinued treatment. Based on the expression of PD-L1, ORR was 7.4% (PD-L1 < 1%) and 6.8% (PD-L1 < 10%) for the dual therapy, and 8.8% (PD-L1 < 1%) and 8.9% (PD-L1 < 10%) in the arm treated with durvalumab. In the HPV-positive patients, ORR was higher in the arm treated with durvalumab (16.7%) [68].

Ferris et al. [69] developed the first phase III clinical trial (EAGLE) evaluating the efficacy of durvalumab alone or combined with tremelimumab compared to standard therapy in 736 patients with R/M HNSCC who progressed after platinum therapy (Table 2). After evaluating the expression of PD-L1 by TC, the results showed that most patients, in all three therapeutic modalities, presented a TC < 25%. It was also verified that the dual therapy was able to promote an antitumour response of 18.2%, with 6 CR and 39 PR, followed by a rate of 17.9% (6 CR and 37 PR) on monotherapy and 17.3% (43 PR) in standard therapy, with a median DOR of 7.4 months, 12.9 months, and 3.7 months, respectively. Regarding median PFS and OS, no statistically significant differences were observed between the three treatment arms. However, the 1-year OS was slightly higher in monotherapy (37%) compared to dual therapy (30.4%) and standard therapy (30.5%). Approximately 57.4% (durvalumab), 61% (dual therapy), and 82.1% (standard therapy) of patients had AE of any degree, in which anemia and hypothyroidism were the most common. For the same groups, grade 3/4 AE occurred in 10.1%, 16.3%, and 24.2% of patients, respectively. Due to AE, 29 patients discontinued treatment and four died. Based on PD-L1 expression, the median OS was higher in both groups that were administered with immunotherapy [69].

In a phase II clinical trial, André et al. [42] evaluated the combination of an anti-NKG2A agent with cetuximab in 31 patients with R/M HNSCC (Table 2). The results revealed that the ORR of the 26 evaluable patients was 31% and all patients presented with PR. Patients were followed for a mean period of 129 days in which median DOR was not achieved, as six PR patients were still undergoing treatment. In total, 93% of patients had grade 1/2 AE associated with monalizumab, although they were easily reversible and manageable. The most common AEs were fatigue, pyrexia, and headache. Considering the AE associated with cetuximab, there were no AE that were exacerbated by monalizumab, based on previous studies [42].

In a single-arm, phase II clinical trial, Massarelli et al. [48] aimed to evaluate the activity of an anti-PD-1 agent in combination with a peptide vaccine targeting HPV-16 oncoproteins E6 and E7 (Table 3). The 24 patients selected for the study had incurable solid tumours and were HPV-16 positive, within which 22 had OPC and the remaining developed anal and cervical cancer. After the treatment, only 8 patients, all with OPC, had the following responses: 8% had a CR and 25% had a PR, resulting in an ORR of 33%. The median duration of the response was 10.3 months. The median PFS and OS of the population under study were 2.7 months and 17.5 months, respectively. Considering the 22 patients with OPC, the results were similar. Regarding safety, most patients experienced expected vaccine-associated effects such as pain and fever at the injection site, fatigue, diarrhoea, and nivolumab-associated hepatotoxicity. Grade 3/4 imAE led to 2 patients discontinuing treatment. For patients with CPS ≥ 1, there was an ORR of 43% parallel to the ORR of 18% in patients with negative PD-L1 expression; however, these results were not related to the response. After vaccination, there was also an increase in the number of HPV-specific T cells in the general population; however, the immune response was not correlated with any efficacy outcome [48].

In 2019, Aggarwal et al. [49] published the results of a phase Ib/II study based on immunotherapy with MEDI0457 in 22 patients with locally advanced HNSCC and HPV-positive (Table 3). At a mean follow-up of 15.9 months, 3 patients developed disease progression, so the 1-year disease-free survival (DFS) rate was 89.4%. After the end of treatment, patients were evaluated for immunological response based on the generation of IgG class antibodies and the presence of peripheral blood mononuclear cells (PBMC) against E6 and E7 antigens. It should be noted that approximately 88.2% and 64.7% of subjects with HPV-18 and HPV-16, respectively, presented higher seroreactivity peaks against the E7 antigen compared to the E6 antigens of both genotypes. At the 6-month follow-up, patients revealed that they had detectable antibody titers against at least one of the four antigens. However, after 23 months, only one patient demonstrated persistent reactivity against any antigen. Regarding PBMC, patients in cohort I presented median increases of 63 SFU/10^6^ PBMC and 75 SFU/10^6^ PBMC for HPV-16 and HPV-18 genotypes, respectively. In cohort II, there were median increases of 75 SFU/10^6^ PBMC and 55 SFU/10^6^ PBMC for the same genotypes. After 3 months of the end of the last dose, the responses decreased; however, remained above the baseline of most patients. In addition, the number of CD8^+^ T cells specific for viral antigens was also evaluated in 8 patients based on surface markers CD38, CD69, and CD137, as well as cytolytic capacity by the presence of granzyme A, granzyme B, and perforins (Prf). Overall, 5 of the 8 patients presented specific CD8^+^ T cell elevations against HPV-16 and HPV-18 in at least one of the parameters analyzed, with the production of granzymes and Prf. All the AE reported were grade 1, with pain at the injection site being the most prevalent [49].

In 2021, Julian et al. [10] reported the results of an open phase Ib/IIa study combining an anti-PD-L1 agent (durvalumab) with MEDI0457 in 27 patients with R/M HNSCC and HPV-positive (Table 3). In the results presented, the ORR was 22.2% with 3 CR and 3 PR. In this study, the induction of HPV-specific peripheral T cells and antibodies was further quantified. Thus, an increase in infiltrating CD8^+^ T cells and peripheral HPV-specific T cells was observed. According to the AE, 77.1% of the patients presented grade 1/2 AE, in which fatigue and pain at the injection site were the most frequent. In addition, 14.3% and 2.9% had grade 3 AE and three grade 3 severe AE (AST and ALT elevations, and myocarditis) [10].

Reuschenbach et al. [47] evaluated the safety and immunogenicity of vaccine P16_37-63, combined with Montanide ISA-51 VG, in 26 patients with advanced cancer and HPV-positive with overexpression of p16^INK4a^, as determined by immunohistochemistry (Table 3). Of these 26 patients, 7 had been diagnosed with HNC. Regarding tumour response, no patient revealed a CR or PR. For patients with HNC, the mean OS and PFS were 11.05 and 6.86 months, respectively. To evaluate immunogenicity, 20 patients were eligible because they had completed at least 1 course of treatment with subsequent immunological evaluation (CD4^+^ and CD8^+^ T cells, and mAb). About 85% presented immunological response against the P16_37-63. In HNC patients, all demonstrated positive immune responses to CD4+ T cells (85%) and/or antibodies (85%) of magnitudes + a +++ during therapy and follow-up 2 months after the end of vaccination. In addition, one patient also showed a positive immune response mediated by CD8^+^ T lymphocytes. As for toxicity, no serious adverse reactions associated with therapy were observed; however, mild reactions in the injection site were observed [47].

Si et al. [12] developed a phase I clinical trial with a viral vector-based vaccine (rAd5-EBV-LMP2) in 24 patients with advanced-stage NPC (Table 3). These patients were evaluated for their safety and immune response based on peripheral CD4^+^ and CD8^+^ T cells. Blood samples were collected before and after the administration of the last dose of the vaccine. Regarding the immunological response, patients submitted to the two lowest doses of the vaccine did not show significant results regarding the proportion of CD4^+^ and CD8^+^ T cells. However, 66.6% of patients submitted to the highest dosage demonstrated a significant increase in the proportion of CD4^+^ T lymphocytes compared to the baseline levels initially quantified. In the other subgroups, only 16.7% and 22.2% of those inoculated with low and medium doses, respectively, demonstrated an increase in the cell level reported. About CD8^+^ T lymphocytes, there was no significant increase in the three dosages under study. After the end of therapy, patients were followed for 2 years, in which only 4 patients had progression of the neoplasm. Regarding the safety of treatment, toxicity related to the inoculation of the vaccine was mild and tolerable (grade 1/2). Among these, the most common were fatigue, myalgia, and cough [12].

In a phase Ib/III multicenter study, Harrington et al. [70] evaluated the antitumour effect of T-VEC in combination with pembrolizumab in 36 patients with platinum-refractory R/M HNSCC (Table 3). When administering T-VEC, patients could be injected at different sites (skin, subcutaneous, and nodal lesions), up to a total of 8 mL. In this study, the ORR was 13.9%, which is all PR. The median DOR was not achieved. Importantly, note that all the responders presented positive PD-L1 expression (CPS ≥ 1). The median PFS and OS were 3.0 and 5.8 months, respectively. During the trial, there was reported one dose-limiting toxicity, including a fatal arterial hemorrhage after the administration of 2 doses of treatment. For AE associated with T-VEC and pembrolizumab, 55.6% and 58.3% of patients had AE of any degree, respectively. However, 19.4% of patients had severe AE from each of the therapies. Because of these, three patients discontinued the treatment [70].

## 4. Discussion

Many of the patients with advanced, recurrent, and/or metastatic disease show difficulties in responding to standard therapy, and ultimately, the disease progresses after a short period of time. Due to the toxicity associated with previous therapeutic modalities and low survival rates, the administration of immunotherapy (ICI, therapeutic vaccines, and oncolytic viruses) in these patients was explored.

Based on the KEYNOTE-012 and CheckMate 141 trials, the first immunotherapies for patients with R/M HNSCC were approved—pembrolizumab and nivolumab [14]. Although the response rates of these mAb remain moderate, both immunotherapies were able to promote an increase in OS and PFS, compared to standard therapy [56,60]. Therefore, one can speculate that the patients that will benefit are not only the responders who benefit from mAB in long term but also the non-responder patients.

Such evidence suggests the therapeutic efficacy of these mAbs, as single agents, in the treatment of patients with poor prognoses and limited therapeutic options.

Pembrolizumab was also involved in two other studies that evaluated its clinical activity in patients with R/M HNSCC (KEYNOTE-055), in addition to comparing it to the standard therapy usually given (KEYNOTE-040). In both studies, pembrolizumab continued to exhibit clinical activity similar to that observed in KEYNOTE-012 [57,58]. In addition, pembrolizumab was able to prolong OS and exhibit a higher safety profile compared to standard therapy administered in patients with R/M HNSCC. Although response rates remain moderate, mAb can delay the durability of the response, thus suggesting its long-term therapeutic effect. Pembrolizumab continued to demonstrate a safety and efficacy profile in pretreated populations exposed to high toxicities [71].

It is known that patients with OC-SCC also have a high recurrence rate and, often, a high risk of death, even in the early stages [61]. In these patients, it is usually common to perform surgical resection followed by adjuvant therapy [61] and, therefore, it will be advantageous to explore the effect of immunotherapy on the neoadjuvant setting. Immunotherapy in this scenario can be able to induce a more improved immune response with the ability to reduce tumour volume, thus facilitating surgical intervention. Furthermore, to further enhance the antitumour response, it has become important to explore the addition of anti-CTLA-4 agents, such as ipilimumab, to anti-PD-1 agents. Although high rates of volumetric response and pathological downstaging are similar in both treatment arms, ipilimumab administered at lower dosages was able to induce a higher ORR compared to nivolumab monotherapy. In addition, these patients exhibit a long-term clinical benefit, with promising PFS and OS rates. For that reason, the administration of nivolumab with ipilimumab resulted in a promising response rate for the treatment of these patients.

In several studies, patients with HPV tend to respond better to immunotherapies with ICI compared to HPV-negative patients. CheckMate 358 was developed to explore the differences between the two classes of HNSCC in the neoadjuvant scenario with the administration of nivolumab [59]. Of the patients submitted to surgery, HPV-positive patients had one MPR and three partial responses, while in HPV-negative patients, only one achieved a partial response [59]. The results were also shown to be similar in biopsied patients. Such results may be associated with the fact that TME in HPV-positive patients is less immunosuppressive and more inflamed compared to HPV-negative patients. Therefore, HPV-positive patients respond earlier to immunotherapy. Although these patients presented better radiographic and pathological rates compared to HPV-negative patients, these rates remained low [59], thus correlating with the possible highly immunosuppressive TME in patients with HNC. The rates of RFS in two years were still promising for both groups under treatment, thus revealing the efficacy of nivolumab in the treatment of these patients, regardless of HPV status [59]. However, it should be considered that the presence of strongly immunogenic viral neoantigens may potentiate greater reactivity on the part of the immune system.

To block another inhibitory molecule, HAWK and CONDOR were developed to explore the administration of durvalumab (anti-PD-L1). Although both studies focus on patients with R/M HNSCC, the studies show differences in the PD-L1 expression in the tumour. Similar to pembrolizumab, durvalumab monotherapy revealed a tumour activity of 16.2% [72]. To assess if the antitumour response could be potentiated by the addition of an anti-CTLA-4 agent, CONDOR also added tremelimumab. No clinical benefits were observed when tremelimumab was added to durvalumab monotherapy, revealing an ORR of only 7.8% [68]. However, it is necessary to consider that these results may have been affected by the tumour expression of PD-L1 being lower than 25%.PFS and OS rates were shown to be similar in both studies without any significant differences [65,68], which emphasizes the idea that durvalumab is a mAb capable of improving the survival of these patients regardless of the expression of PD-L1.

EAGLE was another study that aimed to demonstrate the efficacy of immunotherapy with durvalumab or durvalumab combined with tremelimumab compared to standard therapy. Tremelimumab continued to show no clinical benefit compared to monotherapies. However, immunotherapy administered in both treatment arms revealed greater therapeutic durability compared to standard therapy [69], thus revealing its long-term beneficial effect on the treatment of patients with R/M HNSCC. This study is partly compromised by the greater misunderstood evidence of OS in the group submitted to standard therapy, thus conditioning the therapeutic credibility of durvalumab [69]. Therefore, the effectiveness of durvalumab should be further explored.

To evaluate the effect of PD-L1 blockade, two other studies were developed with anti-PD-L1 agents—avelumab and atezolizumab [66,67]. Both mAbs demonstrated clinical activity with ORR rates of 13.1% and 22%, with considerable median response durations [66,67]. Such evidence has proved similar to previous studies conducted with other inhibitory agents (pembrolizumab, nivolumab, and durvalumab), in which ORR ranged from 13% to 18% [66]. Avelumab and atezolizumab exhibited long-term clinical activity, thus evidencing their clinical efficacy. In addition, the administration of both mAbs was safe for most patients, with no significant adverse events. In this sense, avelumab and atezolizumab seem to be therapeutic options for patients with HNC in advanced stages.

Since radiotherapy and QT are used in patients with HNSCC, it will become advantageous to explore its combination with immunotherapy to understand whether there are clinical benefits. Note that immunogenic cell death induced by both therapies can potentiate the antitumour response [2,73], suggesting that their combination with ICI may further improve this antitumour response.

Mc Bride et al. [62] evaluated the combination of nivolumab and SBRT compared to nivolumab monotherapy in metastatic patients. This strategy would be promising for these patients because radiotherapy can stimulate systemic immunity, caused by the immunogenic death exerted on irradiated tumour cells, thereby inducing consequent antitumour responses in non-irradiated sites, such as metastases. Both therapeutic approaches exhibited an efficient safety profile with a very small percentage of more severe adverse events. Regarding efficacy, there were no clinical benefits in the group submitted to immunotherapy and radiotherapy, and the highest rate of ORR was observed in the group submitted to immunotherapy alone. HNC can affect different areas, so variable immunogenicity between them may have conditioned the response. In addition, the dose, time, and volume of the irradiated lesion may also be associated with these results [62].

Two other studies have been developed combining Sintilimab with induction chemotherapy in the neoadjuvant setting, as well as pembrolizumab with Docetaxel [63,64]. In both studies, the combination of mAb with QT revealed clinically significant ORR rates compared to QT or immunotherapy (KEYNOTE-012, KEYNOTE-040, and KEYNOTE-055) isolation [63,64]. In addition, combination therapies also revealed their efficacy at the level of median OS and PFS compared to QT and studies where pembrolizumab was evaluated in isolated [63,64]. Both therapies exhibited a good safety profile, although monotherapies have safer profiles [63,64]. It should be noted that QT combined with immunotherapy in the neoadjuvant setting proved to be a promising strategy for patients, eligible for platinum, with neoplasia in advanced stages and may eventually enhance the conversion of an unresectable tumour into resectable and reduce tumour volume, thus facilitating surgery. Although these strategies have proved promising, further studies should be developed to improve the clinical benefits of combining QT with immunotherapy.

Because response rates remain low when the PD-1/PD-L1 pathway is blocked, it has become important to explore other inhibition pathways—NKG2A. With the inhibition of NKG2A with monalizumab, an increase in the antitumour response mediated by NK and CD8^+^ T cells is expected. It is important to note that the combination of this mAb with cetuximab was able to induce an ORR of 31% and a manageable safety profile [42]. Such evidence contrasts with other ICI monotherapy, suggesting that monalizumab was able to stimulate the antitumour response mediated by cytotoxic immune cells. Although median response durability and maximum tolerated dose have not been achieved [42], previous evidence is promising for future studies in HNC patients.

In addition to ICI, immunotherapy has been developing new approaches in the HNSCC scenario—therapeutic vaccines and oncolytic viruses. Given the low infiltration of APC cells and the polarisation of M2 macrophages, innate and adaptive immune activation are compromised [74]. To date, vaccines targeting HPV are the most clinically interesting due to antigenic targets recognised as non-self-formed by the chronic infection.

The combination of ISA-101 with nivolumab revealed a considerable tumour response rate in patients with OPC, as well as an encouraging long-term PFS and OS for these patients [11,48]. This therapeutic approach also promoted the elevation of HPV-specific T cells in responding and non-responding patients to therapy [48]. Such evidence suggests that the administration of this peptide vaccine with nivolumab is capable of potentiating adaptive immunity, which is important for the control of the tumour. Therefore, the safety and efficacy profile of this study highlights the possible effect of the combination of different immunotherapies on the treatment of patients with HPV-associated HNC. In addition, this therapeutic strategy proved even more promising compared to other studies with ICI, administered based on HPV status—CheckMate 141, KEYNOTE-012, and KEYNOTE-055.

Similar to ISA-101, MEDI0457 was also well tolerated by the study population without a record of severe AE [10,49]. The combination of the DNA vaccine with durvalumab revealed an ORR (22.2%) higher [10], compared to studies in which mAb was administered in isolation—HAWK, CONDOR, and EAGLE. These results clearly show the beneficial effect of increased antigenic presentation by vaccine administration coupled with PD-L1 inhibition. Therefore, therapeutic vaccines can modify the TME of these patients, inducing a necessary pro-inflammatory state in the antitumour response [49]. In addition, it was also found that the vaccine was sufficient to induce tumour-specific CD8^+^ T-cell immunity and HPV-specific peripheral T cells [10,49], which may promote tumour regression in these patients. The isolated administration of MEDI0457 was able to induce humoral immunity, of long persistence, against at least one viral antigen [49]. Thus, the nucleic acid vaccine was able to stimulate a specific and lasting immune response against tumour-specific antigens, suggesting that its combination with ICI may increase the efficacy of the antitumour response, thereby controlling the neoplasm and reducing the risk of new lesions [49,75].

P16 is a protein commonly overexpressed in patients with HPV-associated HNC. Although P16_37-63 did not demonstrate a pathological response, all patients revealed cellular immunity, mediated by CD4^+^ and CD8^+^ T lymphocytes, and humoral [47,75]. In some cases, a vaccine dose was sufficiently capable of inducing the same immune response [47], important in the control of neoplasm. In this sense, patients with HNSCC in advanced stages and with overexpression of P16 may benefit from this vaccine; however, this study should be administered to a wider population to understand its real antitumour effect.

The Epstein–Barr virus infection may also be associated with the development of HNC at the NPC. The rAd5-EBV-LMP2 vaccine exhibited an efficient safety profile as well as cellular immunity [12]. Upon administration of the vaccine, it was found that only the highest dosage induces greater cellular immunity, essentially at the level of peripheral CD4^+^ T lymphocytes [12]. It should be noted that the low immunogenicity of this vaccine may have conditioned the stimulation of the immune response mediated by CD8^+^ T cells. In this sense, additional studies should be developed to amplify, in the near future, the therapies in patients with HNC associated with viral infections.

Finally, oncolytic viruses, such as T-VEC, can selectively infect and replicate in tumour cells, inducing their lysis and stimulating a specific immune response. Therefore, its combination with pembrolizumab would improve the antitumour response. This combination demonstrated an efficient safety profile since most adverse events presented were mild [70]. However, the combination of these two immunotherapies did not seem to improve ORR (13.9%) [70] compared to pembrolizumab monotherapy in other studies—KEYNOTE-012, KEYNOTE-055, and KEYNOTE-040. Thus, T-VEC was not able to modify the TME of these patients to stimulate the antitumour response. However, the effectiveness of this therapy may have been compromised by the early death of enrolled patients [70] and, for this reason, additional studies should be developed to explore the possible improved antitumour response that T-VEC may induce.

It should be interesting to note that, in most studies, patients with HPV-associated HNC and/or elevated expressions of PD-L1 tend to demonstrate a greater clinical benefit when administering immunotherapies, as well as a better prognosis. Patients with HPV-associated HNC have a highly inflamed and less immunosuppressive TME, with high infiltration of lymphocytes with low PD-1 expression [10,25]. For this reason, these patients end up responding earlier to immunotherapy, thus demonstrating better clinical benefits. On the other hand, PD-L1 is characterised by being a dynamic marker, as its expression is dictated by an inflammatory TME [15,18,23,36,38]. Although patients with reduced expressions of PD-L1 also respond to immunotherapy, its higher expression is often correlated with greater effectiveness of therapy. For this reason, these biomarkers must remain in the investigation, as their presence is predictive of the greater effectiveness of immunotherapy. In this way, such evidence may contribute to a more adequate therapeutic selection based on the characteristics of each patient. In Figure 4, we can observe the different immunotherapeutic approaches used in HNSCC patients’ treatment.

## 5. Conclusions

When diagnosed, patients with HNC are already in the advanced stages of cancer with a high probability of recurrence and the development of metastases. For these patients, therapeutic options are difficult and limited. Due to the high morbidity and toxicity, immunotherapy was an important therapeutic option. Recognizing the different subtypes of HNC will become relevant to selecting a therapy, since molecular and immunological differences, as well as differences in tumour heterogeneity, may condition the answer to immunotherapy.

In this scenario, HPV-directed therapeutic vaccines have become a strategy on the rise, given their ability to induce a durable cellular and humoral immune response. Therefore, these represent a possible therapeutic option for HPV-positive patients, as monotherapy or combined with other agents to potentiate the antitumour response. The combination of T-VEC with pembrolizumab did not enhance the antitumour response in patients with R/M HNSCC, thus contrasting with results obtained in other tumours.

However, monotherapy with ICI is the most impactful strategy in this neoplasm scenario due to its ability to induce a durable antitumour response, prolonging the survival of patients with locally advanced, recurrent, or metastatic disease in the face of standard therapy. In addition, this strategy was also correlated with a higher quality of life and a lower toxicity rate. Although the combination of anti-PD-L1 agents with anti-CTLA-4 reveals an increase in antitumour immunity, patients with R/M HNSCC do not seem to benefit from this strategy. However, in the neoadjuvant setting, the addition of ipilimumab to an anti-PD-1 agent showed promising results in patients with untreated OC-SCC. When combining ICI with QT, in different scenarios, patients tend to reveal higher response rates and long-term clinical benefits, in contrast to the results obtained when combining ICI with radiotherapy. Although it has proved to be an advantageous strategy for these patients, the toxicity associated with therapy is still a barrier to overcome.

Even though therapy with ICI has revealed clinical benefits for these patients, ORR rates still remain moderate. In this sense, it will become essential to recognise the mechanisms of tumour resistance to optimise therapies.

## Figures and Tables

**Figure 1 biomedicines-10-02151-f001:**
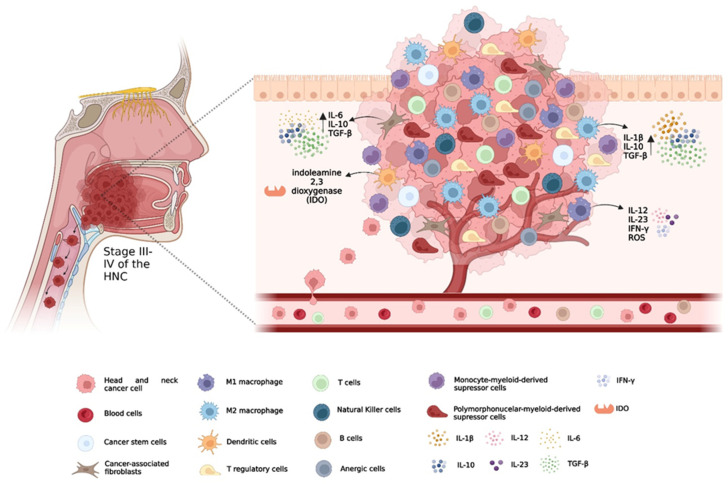
The tumour microenvironment (TME) in head and neck cancer: Cancer-associated fibroblasts perform vital functions in angiogenesis, tumour invasion and metastasis, and inhibit T-cell function by cytokine secretion. The presence of vascular endothelial growth factor, interleukin (IL) 6 and granulocyte-macrophage colony-stimulating factor in the TME promote the aggregation of myeloid-derived suppressor cells, with the ability to suppress the activity of T cells from the production of nitric oxide by nitric oxide synthase. The presence of M2 macrophages is correlated with a worse prognosis, given their ability to release immunosuppressive cytokines and negatively regulate the activity of M1 macrophages. Effector cells and antigen presenting cells (APC) see their functions compromised from the release of immunosuppressive cytokines, B granzymes and perforins by regulatory T cells (T reg). In addition, T reg regulate the production of indoleamine-2,3-dioxygenase 1 by APC, suppressing the essential tryptophan in the proliferation of T cells, in addition to inhibiting antigenic presentation by the expression of inhibitory molecules. The presence of T and Natural Killer cells aim to exert an antitumour response; however, their function is impaired by a highly immunosuppressive TME.

**Figure 2 biomedicines-10-02151-f002:**
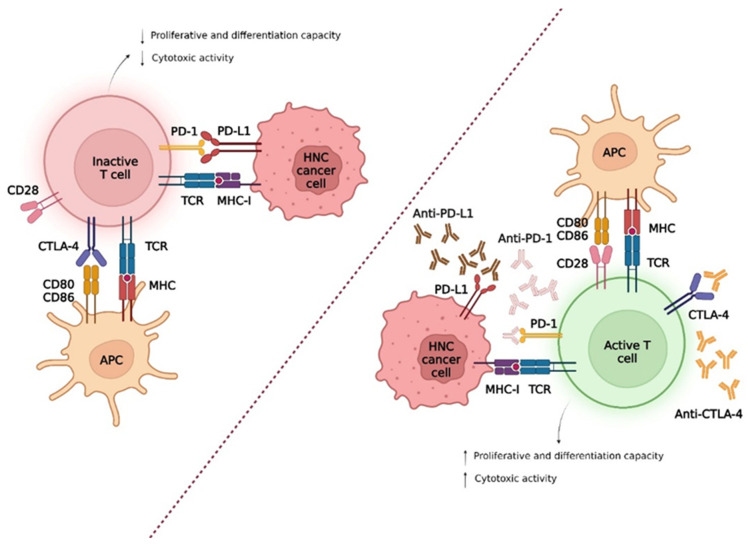
Tumour evasion mechanisms, in patients with head and neck cancer, from immunological checkpoints. The interaction between programmed death-1 (PD-1), expressed in T cells, with the programmed death-ligand 1 (PD-L1) promotes the inactivation of T cells and consequent proliferation. The inactivation of T cells mediated by cytotoxic T lymphocyte antigen 4 (CTLA-4) occurs because CTLA-4 has a higher affinity for CD80/CD86 ligands compared to CD28. In this sense, the development of anti-PD-1/PD-L1 and anti-CTLA-4 antibodies prevent the occurrence of these interactions, thus allowing the reestablishment of antitumour immunity mediated by T cells.

**Figure 3 biomedicines-10-02151-f003:**
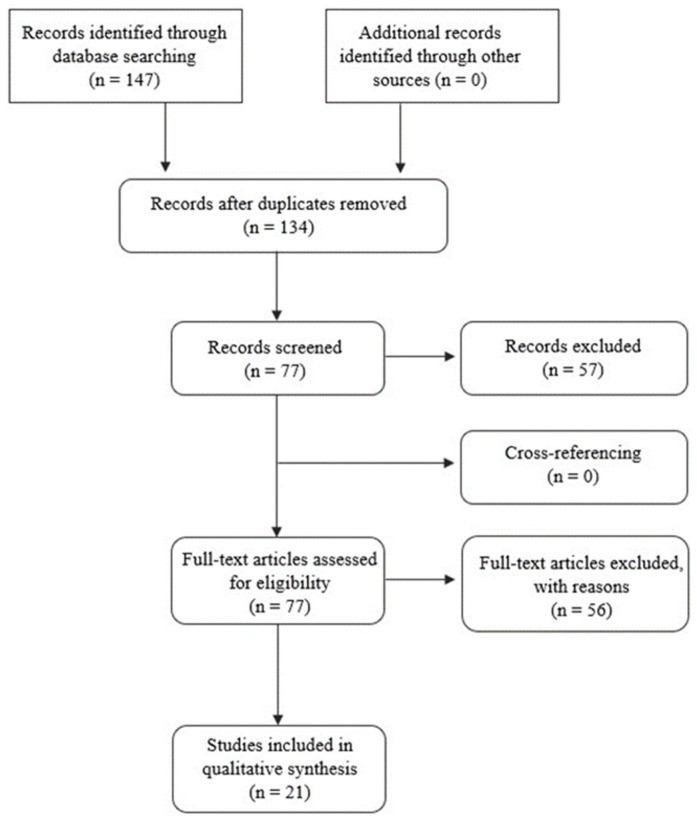
Prism of the systematic review, where the literature searches in the database are detailed, the number of articles selected based on the abstract, the number of articles evaluated by the eligibility criteria, and the total number of articles included in the review.

**Figure 4 biomedicines-10-02151-f004:**
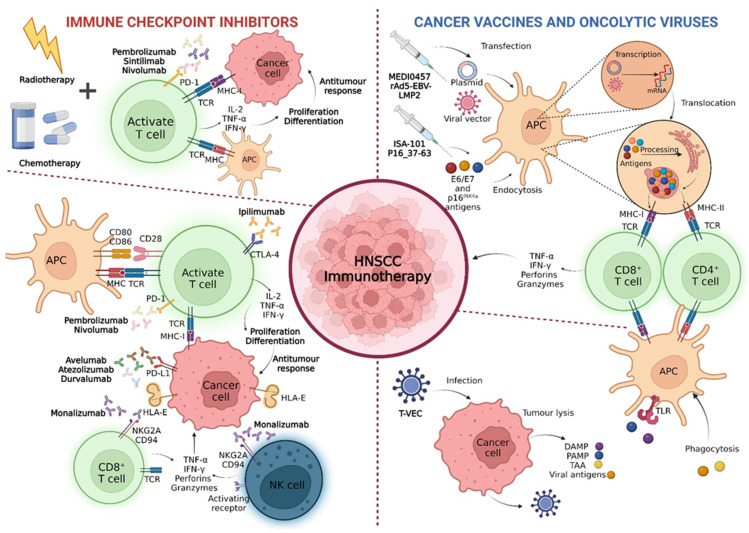
Immunotherapy in the treatment of patients with HNC. To prevent the development of autoimmune diseases due to the hyperreactivity of the immune system, T cells express inhibitor molecules on the surface: programmed death-1 (PD-1) and cytotoxic T lymphocyte antigen 4 (CTLA-4). From 50% to 60% of patients with head and neck squamous cell carcinoma express the PD-1 ligand in their tumour cells—programmed death ligand 1 (PD-L1). Upon this interaction, T cells suppress their cytotoxic activity and, consequently, promote tumour progression. Likewise, the interaction of CTLA-4 with the CD80/CD86 ligands expressed on antigen presenting cells (APC) prevents the proliferation and differentiation of T cells. NK group 2 member A (NKG2A) is a molecule expressed on natural killer and CD8^+^ T cells that, after interacting with HLA class I histocompatibility antigen, alpha chain E (HLA-E) present in tumour cells, recruits phosphatases to induce inhibitory signals to immune cells. In this sense, immunotherapy with antibodies directed to these molecules aims to reestablish the antitumour immunity in the same way as to promote tumour regression. Chemotherapy and radiotherapy can induce immunogenic cell death, and consequent activation of the adaptive immune response. Therefore, its combination with anti-PD-1 agents aims to improve the immune response. In addition to these strategies, therapeutic vaccines targeting Human Papilloma Virus and Epstein–Barr, and oncolytic viruses are other immunotherapeutic approaches on the rise. Nucleic acid vaccines, viral vector-based vaccines, and peptide vaccines integrate plasmids, viral vector, and viral antigens, respectively, into APC. In this way, the antigens will be processed and presented in the major histocompatibility complex (MHC) context, on the cell surface, stimulating cellular and humoral immunity. Talimogene laherparepvec (T-VEC), after infecting tumour cells from specific receptors, can replicate and induce lysis of these same cells. In this way, damage-associated molecular patterns (DAMP) and pathogens-associated molecular patterns (PAMP), as well as tumour-associated antigens (TAA) and viral antigens, are released into the medium. After their recognition by APC cells, immunity mediated by T CD4^+^ and T CD8^+^ cells are stimulated.

**Table 1 biomedicines-10-02151-t001:** Strategy used for the combination of keywords (Medical Subject Headings terms) to obtain the literature used in this review.

Medical Subject Headings Terms	And	Or	Not
Head and neck neoplasm	Therapeutics		
Head and neck neoplasm	Immune Checkpoint Inhibitors		
Head and neck neoplasm	Therapeutics	Immune Checkpoint Inhibitors	
Head and neck neoplasm	Immune Checkpoint Inhibitors		Cancer Vaccines, Oncolytic Viruses
Head and neck neoplasm	Cancer Vaccines		Immune Checkpoint Inhibitors, Oncolytic Viruses
Head and neck neoplasm	Oncolytic Viruses		Immune Checkpoint Inhibitors, Cancer Vaccines
Head and neck neoplasm	Immune Checkpoint Inhibitors, Cancer Vaccines		Oncolytic Viruses
Head and neck neoplasm	Immune Checkpoint Inhibitors, Oncolytic Viruses		Cancer Vaccines

**Table 2 biomedicines-10-02151-t002:** Recent studies of immune checkpoint inhibitors, such as pembrolizumab, nivolumab, durvalumab, atezolizumab, avelumab, and monalizumab, as monotherapy, and their combinations with anti-CTLA-4 agents, chemotherapy, and radiotherapy.

Study	Antigen Target	Drug	Methods	Results
Mehra R. et al., 2018/KEYNOTE-012 [56]	**PD-1**	Pembrolizumab	192 patients with R/M HNSCC received different doses of Pembrolizumab: 10 mg/kg every 2 weeks (initial cohort) vs. 200 mg every 3 weeks (expansion cohort); Determination of PD-L1 by IHC.	ORR was 18%: 8 CR and 26 PR;
				1-year PFS and OS was 17% and 38%;
				13% of patients present with grade 3/4 AE;
				**HPV-positive patients vs. HPV-negative patients:**
				ORR was 24% vs. 16%;
				**PD-L1 expression (CPS ≥ 1 vs. CPS < 1):**
				ORR was 21% vs. 6%; Median PFS and OS were 2.1 months and 10 months vs. 2 months and 5 months.
Bauml J. et al., 2017/KEYNOTE-055 [57]	**PD-1**	Pembrolizumab	171 patients with R/M HNSCC received 200 mg of Pembrolizumab every 3 weeks; Determination of PD-L1 and HPV status by IHC.	ORR was 16%: 1 CR and 27 PR;
				Median DOR of 8 months;
				6-months PFS and OS was 23% and 59%;
				15% of patients had grade ≥ 3 AE;
				**HPV-positive patients vs. HPV-negative patients:**
				ORR was 16% vs. 15%; 6-months PFS was 25% vs. 21%; 6-months OS was 72% vs. 55%;
				**PD-L1 expression (CPS ≥ 1 vs. CPS < 1):**
				ORR was 18% vs. 12%; 6-months PFS was 24% vs. 20%; 6-months OS was 59% vs. 56%.
Cohen E. et al., 2019/KEYNOTE-040 [58]	**PD-1**	Pembrolizumab	495 patients stratified into 2 distinct arms: Pembrolizumab: 200 mg every 3 weeks; SoC: 40 mg/m^2^ of Methotrexate, weekly; 75 mg/m^2^ of Docetaxel, every 3 weeks; 250 mg/m^2^ of Cetuximab weekly; Determination of PD-L1 and P16 by IHC.	ORR was 14.6% (P) vs. 10.1% (SoC);
				Median PFS was 2.1 months (P) vs. 2.3 months (SoC);
				1-year OS was 37% (P) vs. 26.5% (SoC);
				13% (P) and 36% (SoC) of patients had grade 3/4 AE;
				**PD-L1 expression (CPS < 1 vs. CPS ≥ 1):**
				Median OS was 6.3 months (P) and 7.0 months (SoC) vs. 8.7 months (P) and 7.1 months (Soc);
				**PD-L1 expression (TPS < 50% vs. TPS ≥ 50%):**
				Median OS was 6.5 months (P) and 7.1 months (SoC) vs. 11.6 months (P) and 6.6 months (SoC).
Ferris RL. et al., 2021/CheckMate 358 [59]	**PD-1**	Nivolumab	52 patients with HNSCC received 240 mg of Nivolumab on days 1 and 15 in the neoadjuvant setting; Determination of PD-L1 by IHC; Determination of P16 by IHC, FISH or PCR.	**HPV-positive patients vs. HPV-negative patients:**
				Pathological response rate was 23.5% vs. 5.9%;
				2-years RFS was 88.2% vs. 52.2%;
				3-years OS was 100% vs. 63.5%;
				19.2% vs. 11.5% of patients had grade 3/4 AE.
Ferris RL. et al., 2020/CheckMate 141 [60]	**PD-1**	Nivolumab	361 patients with R/M HNSCC stratified into 2 distinct arms: Nivolumab: 3 mg/kg every 2 weeks; SoC: 40–60 mg/m^2^ of Methotrexate + 30–40 mg/m^2^ of Docetaxel + 250 mg/m^2^ of Cetuximab weekly; Determination of PD-L1 and P16 status by IHC.	ORR was 13.3% (N) vs. 5.8% (SoC);
				6-months PFS was 19.7% (N) vs. 9.9% (SoC);
				1-year OS was 36% (N) vs. 16.6% (SoC);
				13.1% (N) and 35.1% (SoC) of patients had grade 3/4 AE;
				**HPV** **-positive patients vs. HPV-negative patients:**
				Median OS was 9.1 months (N) and 4.4 months (SoC) vs. 7.5 months (N) and 5.8 months (SoC).
Schoenfeld JD. et al., 2020 [61]	**PD-1 and CTLA-4**	Nivolumab and Iplimumab vs. Nivolumab	29 patients with OC-SCC stratified into 2 different arms: Nivolumab + Ipilimumab: 3 mg/kg of N (weeks 1 and 3) and 1 mg/kg of I (week 1); Nivolumab: 3 mg/kg at weeks 1 and 3; Therapy administered in the neoadjuvant setting.	ORR was 38% (N + I) vs. 13% (N);
				Volumetric response was 53% (N + I) vs. 50% (N);
				Pathological downstaging was 53% (N+I) vs. 69% (N);
				1-year PFS and OS was 85% and 89%;
				5 (N + I) and 2 (N) patients had grade 3/4 AE.
McBride S. et al., 2021 [62]	**PD-1**	Nivolumab and SBRT vs. Nivolumab	62 patients stratified into 2 different arms: Nivolumab + SBRT: 3 mg/kg N, every 2 weeks, and SBRT (9 Gy × 3); Nivolumab: 3 mg/kg every 2 weeks; Determination of PD-L1 by IHC.	ORR was 34.5% (N) vs. 29% (N + SBRT);
				Median DOR was 9.4 months (N + SBRT);
				1-year PFS was 32.2% (N) vs. 16.8% (N + SBRT);
				1-year OS was 50.2% (N) vs. 54.4% (N + SBRT);
				13.3% (N) and 9.7% (N + SBRT) of patients had grade 3-5 AE;
				**HPV-positive patients vs. HPV-negative patients:**
				ORR was 41.9% vs. 20.7%; 1-year PFS was 64.4% vs. 40.5%;
				**PD-L1 expression (TC < 1% vs. TC ≥ 1%):**
				ORR was 23.5% vs. 50%; 1-year PFS was 47% vs. 63%.
Li X. et al., 2021 [63]	**PD-1**	Sintilimab and IC vs. IC	163 patients with locally advanced HNSCC stratified into 2 distinct arms: IC: Docetaxel (75 mg/m^2^), Platinum (75 mg/m^2^) and Fluorouracil (750 mg/m^2^/day for 5 days) for 2 cycles; Sintilimab + IC: 200 mg on day 1 of each cycle, every 3 weeks + IC.	ORR was 68.4% (IC) vs. 84.6% (Sintilimab + IC);
				2-years OS was 61% (IC) vs. 70% (Sintilimab + IC);
				2-years PFS was 27% (IC) vs. 44% (Sintilimab + IC);
				15.3% (IC) and 18.5% (Sintilimab + IC) of patients had grade 3/4 AE.
Fuereder T. et al., 2022 [64]	**PD-1**	Pembrolizumab and Docetaxel	22 patients received 75 mg/m^2^ DTX and 200 mg Pembrolizumab every 3 weeks for 6 cycles, followed by maintenance therapy with Pembrolizumab every 3 weeks.	3-months ORR was 22.7%: 1 CR and 4 PR;
				1-year OS was 68.2%;
				1-year PFS was 27.3%;
				13.6% of patients had grade 3 AE.
Zandber D. et al., 2019/HAWK [65]	**PD-L1**	Durvalumab	112 patients with R/M HNSCC received 10 mg/kg of Durvalumab every 2 weeks for 1 year; Determination of PD-L1 (TC ≥ 25%) by IHC; Determination of P16 by IHC, FISH or PCR.	ORR was 16.2%: 1 CR and 17 PR;
				1-year PFS was 14.6%;
				1-year OS was 33.6%;
				9 patients had grade 3/4 AE;
				**HPV-positive patients vs. HPV-negative patients:**
				ORR was 29.4% vs. 10.8%; Median PFS and OS were 3.6 months and 10.2 months vs. 1.8 months and 5.0 months.
Colevas AD. et al., 2018 [66]	**PD-L1**	Atezolizumab	32 patients with HNC received 15 mg/kg, 20 mg/kg or fixed dose of 1200 mg every 3 weeks (16 cycles) of Atezolizumab; Determination of PD-L1 by IHC; Determination of HPV status by PCR.	ORR was 22%: all PR;
				Median DOR was 7.4 months;
				Median PFS was 2.6 months;
				1-year OS was 36%;
				13% of patients had grade 3/4 AE;
				**PD-L1 expression: ICs_0/1_ vs. ICs_2/3_**
				ORR was 14% vs. 24%; Median DOR was 7.4 months vs. 26.2 months;
Guigay J. et al., 2021/JAVELIN Solid Tumour Trial [67]	**PD-L1**	Avelumab	153 patients with R/M HNSCC received 10 mg/kg of Avelumab every 2 weeks; Tumour assessment carried out by the IRC and investigator; Determination of PD-L1 and HPV status by IHC,	ORR was 9.2% (IRC) vs. 13.1% (investigator);
				Median DOR was 30.4 months (investigator);
				1-year PFS was 10.7% (IRC) vs. 13.5% (investigator);
				1-year OS was 35.9%;
				6.5% of patients had AE grade ≥ 3;
				**HPV-positive patients vs. HPV-negative patients:**
				ORR was 15.4% (IRC) and 17.9% (investigator) vs. 5.1% (IRC) and 11.1% (investigator); Median PFS was 2.7 months (IRC) and 3.3 months (investigator) vs. 1.4 months (IRC) and 1.4 months (investigator); Median OS was 11.8 months vs. 7.4 months;
				**PD-L1 expression (TC < 1% vs. TC ≥ 1%):**
				ORR was 3.3% (IRC) and 6.7% (investigator) vs. 10.3% (IRC) and 15% (investigator); Median PFS was 1.4 months (IRC) and 1.5 months (investigator) vs. 1.4 months (IRC) and 1.8 months (investigator); Median OS was 8.9 months vs. 7.9 months.
**Siu L. et al., 2019/CONDOR** [68]	**PD-L1 and CTLA-4**	Durvalumab and Tremelimumab	267 patients with R/M HNSCC stratified into 3 distinct arms: D + T: 20 mg/kg of D + 1 mg/kg of T, every 4 weeks (4 cycles), followed by 10 mg/kg of D, every 2 weeks; D therapy: 10 mg/kg of D every 2 weeks; T therapy: 10 mg/kg of T, every 4 weeks, followed by 2 doses every 12 weeks; Determination of PD-L1 (TC < 25%) by IHC.	ORR was 7.8% (D + T) vs. 9.2% (D) vs. 1.6% (T): all PR;
				Median DOR was 9.4 months (D + T);
				6-months PFS was 13.7% (D + T) vs. 20% (D) vs. 1.9% (T);
				1-year OS was 37% (D + T) vs. 36% (D) vs. 24% (T);
				15.8% (D + T), 12.3% (D) and 16.9% (T) of patients had grade 3/4 AE;
				**PD-L1 expression (TC < 1% vs. TC < 10%):**
				ORR was 7.4% vs. 6.8 (D+T); ORR was 8.8% vs. 8.9% (D);
				**HPV-positive patients:**
				ORR was 5.4% (D + T) vs. 16.7% (D).
Ferris RL. et al., 2020/EAGLE [69]	**PD-L1 and CTLA-4**	Durvalumab and Tremelimumab vs. Durvalumab vs. SoC	736 patients with R/M HNSCC stratified into 3 distinct arms: D: 10 mg/kg every 2 weeks; D + T: 20 mg/kg D + 1 mg/kg T, every 4 weeks (4 doses), followed by 10 mg/kg D, every 2 weeks; SoC: Cetuximab, Taxane, Methotrexate, or a Fluoropyrimidine; Determination of PD-L1 by IHC; Determination of HPV status by IHC, FISH or PCR.	ORR was 18.2% (D+T) vs. 17.9% (D) vs. 17.3% (SoC);
				Median DOR was 7.4 months (D+T) vs. 12.9 months (D) vs. 3.7 months (SoC);
				Median PFS was 2.0 months (D + T) vs. 2.1 months (D) vs. 3.7 months (SoC);
				1-year OS was 30.4% (D + T) vs. 37% (D) vs. 30.5% (SoC);
				16.3% (D + T), 10.1% (D) and 24.2% (SoC) of patients had grade 3/4 AE.
André P. et al., 2018 [42]	**NKG2A and EGFR**	**Monalizumab and Cetuximab**	31 patients with R/M HNSCC received 5 doses of Monalizumab (0.4, 1, 2, 4, and 10 mg/kg) every 2 weeks and 400 mg/m^2^ followed by 250 mg/m^2^ of Cetuximab weekly.	ORR was 31%;
				Median DOR was not reached;
				93% of patients had grade 1/2 AE.

**Legend:** PD-1: programmed death-1; PD-L1: programmed death-ligand 1; CTLA-4: cytotoxic T lymphocyte antigen 4; NKG2A: NK group 2 member A; EGFR: epidermal growth factor receptor; HNC: head and neck cancer; R/M HNSCC: squamous cell carcinoma of the head and neck recurrent/metastatic; OC-SCC: squamous cell carcinoma in the oral cavity; HPV: Human Papilloma Virus; TC: tumour cells; ICs: immune cells; IRC: blinded independent review committee; P: pembrolizumab; D: durvalumab; T: tremelimumab; I: ipilimumab; N: nivolumab; IC: induction chemotherapy; DTX: docetaxel; SoC: standard of care; SBRT: stereotactic body radiation therapy; Gy: gray; ORR: objective response rate; CR: complete response; PR: partial response; DOR: duration of response; PFS: progression-free survival; RFS: recurrence-free survival; OS: overall survival; AE: adverse events; CPS: combined positive score; TPS: tumour proportion score; IHC: immunohistochemistry; FISH: fluorescence in situ hybridization; PCR: polymerase chain reaction.

**Table 3 biomedicines-10-02151-t003:** Recent studies of therapeutic vaccines, targeting human papilloma viruses and Epstein–Barr, and oncolytic viruses, such as T-VEC.

Study	Antigen Target	Drug	Methods	Results
Massarelli E. et al., 2019 [48]	**PD-1 and E6/E7 oncoproteins**	Nivolumab and ISA-101	24 patients with solid tumours (22 with OPC) and HPV-16 positives received 100 µg/peptide, on 3 different days, combined with 3 mg/kg of Nivolumab, every 2 weeks; Determination of PD-L1 by IHC.	ORR was 33%;
				Median DOR was 10.3 months;
				6-months PFS and OS was 37% and 75%;
				2 patients had grade 3/4 immunological AE;
Aggarwal C. et al., 2019 [49]	**E6 and E7 oncoproteins of the HPV-16/18 genotypes**	MEDI0457	22 patients with HNSCC stratified into 2 different groups for administration of 4 doses of vaccine: Cohort I: administered with 1 or 2 doses of the vaccine, in the neoadjuvant setting, completing the 4 doses in the adjuvant setting; Cohort II: administered with 4 doses of vaccine after chemoradiation therapy; Determination of PD-L1 by IHC.	1-year DFS was 89.4%;
				88.2% (HPV-18) and 64.7% (HPV-16) of patients showed superior seroreactivity against E7;
				**Median increases in PBMC (Cohort I vs. Cohort II):**
				63 SFU/10^6^ (HPV-16) and 75 SFU/10^6^ (HPV-18) vs. 75 SFU/10^6^ (HPV-16) and 55 SFU/106 (HPV-18);
				5 patients had specific CD^8+^ T cell elevations against both genotypes;
				21 patients had grade 1 AE.
Julian R. et al., 2021 [10]	**PD-L1 and E6/E7 oncoproteins of the HPV-16/18 genotypes**	Durvalumab and MEDI0457	27 patients with R/M HNSCC and HPV-positive received 7 mg vaccine + 1500 mg Durvalumab.	ORR was 22.2%;
				Elevations of infiltrating CD^8+^ T cells and HPV-specific peripheral T cells;
				17.2% of patients had grade 3 AE.
Reuschenbach M. et al., 2016 [47]	**p16^INK4a^**	P16_37-63 and Montanidade ISA 51 VG	26 patients with advanced solid tumours, HPV-positive and overexpression of p16^INK4a^ received 100 µg/peptide + adjuvant, in 3 cycles; Determination of HPV status by PCR and hybridization; Determination of p16^INK4^ by IHC.	**Patients with HNC:**
				75% and 25% of patients had SD and PD, respectively;
				Mean OS was 11.05 months;
				Mean PFS was 6.86 months;
				85% of patients had a humoral and cellular response.
Si Y. et al., 2016 [12]	**EBV-LMP2**	rAD5-EBV-LMP2	24 patients with NPC stratified into 3 distinct groups given 2 × 10^9^ vp vs. 2 × 10^10^ vp vs. 2 × 10^11^ vp of the vaccine.	66.6% (2 × 10^11^ vp) vs. 16.7% (2 × 10^9^ vp) vs. 22.2% (2 × 10^10^ vp) of patients had an elevation of CD^4+^ T cells;
				11.1% (2 × 10^11^ vp) of patients had CD^8+^ T cell elevations.
Harrington KJ. et al., 2020/MASTERKEY-232 [70]	**PD-1**	T-VEC and Pembrolizumab	36 patients with R/M HNSCC received an initial dose of T-VEC (10^6^ PFU/mL) followed by a dose of 10^8^ PFU/mL every 3 weeks + 200 mg Pembrolizumab every 3 weeks.	ORR was 13.9%;
				DOR was not achieved;
				Median OS was 5.8 months;
				Median PFS was 3.0 months;
				DLT observed in 1 patient;
				19.4% (T-VEC) and 19.4% (mAb) of patients had severe AE.

**Legend:** PD-1: programmed death-1; PD-L1: programmed death-ligand 1; mAb: monoclonal antibodies; HNC: head and neck cancer; R/M HNSCC: squamous cell carcinoma of the head and neck recurrent/metastatic; OPC: oropharynx cancer; NPC: nasopharynx cancer; HPV: Human Papilloma Virus; EBV-LMP2: Epstein–Barr virus latent membrane protein 2; T-VEC: Talimogene laherparepvec; PBMC: peripheral blood mononuclear cells; ORR: objective response rate; SD: stable disease; PD: progression disease; DFS: disease-free survival; DOR: duration of response; PFS: progression-free survival; OS: overall survival; DLT: dose-limiting toxicity; PFU: plaque-forming unit; AE: adverse events; IHC: immunohistochemistry; PCR: polymerase chain reaction.

## Data Availability

Not applicable.

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
