# Peer review of "Immunotherapy in Head and Neck Cancer When, How, and Why?"

_biomedicines, 2022, doi:10.3390/biomedicines10092151_

Round 1

Reviewer 1 Report

Congratulations to Authors for this very interesting manuscript. It shows a good appropriateness of topic for journal, good research design and methodological soundness. The statistics are appropriate for the study. The findings are significant. Discussion and conclusions are clear. 

About references, the Authors should check and correct them for completeness and correctness according to the Biomedicines reference list and citation style guide (initials, journal abbreviation, position of the year ….)

Acronyms should be defined the first time they appear in each of three sections: the abstract; the main text; the first figure or table. When defined for the first time, the acronym should be added in parentheses after the written-out form.

In the text, I have a few comments to Authors to improve the manuscript:

1)    Pag. 5, line 184: Atezolimumab > Atezolizumab

2)    Pag. 5, line 191: TCR > put the definition and after (TCR) 

3)    Pag. 7, line 275: in in vitro assay > better > in an in vitro assay

4)    Pag. 14, line 434: RECIST > put the definition and after (RECIST)

Author Response

“Immunotherapy in Head and Neck Cancer When, How and Why?”

Authors: Daniela Pereira, Diana Martins and Fernando Mendes

First, we would like to thank both reviewers for their coloring to allow this article to be published in the best possible way.

We appreciate all comments and suggestions for improvement.

Below are all the details of the reviews with the answers to each of the suggestions made.

Reviewers Report 1

The research team would like to thank you for corrections and suggestions for improvement to the article.

The bibliography was again revised and corrected according to the style indicated by Biomedicines. In addition, all acronyms were revised again, making the indicated corrections. All acronyms were then described in full, followed by the acronyms so that they could be used throughout the text when necessary.

Finally, minor comments were accepted and corrected.

All changes are highlighted in yellow in the manuscript.

Congratulations to Authors for this very interesting manuscript. It shows a good appropriateness of topic for journal, good research design and methodological soundness. The statistics are appropriate for the study. The findings are significant. Discussion and conclusions are clear. 

About references, the Authors should check and correct them for completeness and correctness according to the Biomedicines reference list and citation style guide (initials, journal abbreviation, position of the year ….)

Acronyms should be defined the first time they appear in each of three sections: the abstract; the main text; the first figure or table. When defined for the first time, the acronym should be added in parentheses after the written-out form.

In the text, I have a few comments to Authors to improve the manuscript:

1)    Pag. 5, line 184: Atezolimumab > Atezolizumab - Changed

2)    Pag. 5, line 191: TCR > put the definition and after (TCR) -  T Cell receptor inserted

3)    Pag. 7, line 275: in in vitro assay > better > in an in vitro assay - corrected

4)    Pag. 14, line 434: RECIST > put the definition and after (RECIST) Response Evaluation Criteria in Solid Tumours inserted

Reviewer 2 Report

While this review has certain value, there are several aspects that need careful attention for the paper to be considered for publication. The Methodology must be revised, as it is not clear what the authors mean by the fact that ‘only freely available papers’ were considered in this article. Also, the search criteria included review articles. Based on this, the authors must clearly state the novelty of their review. Why is this any different from the previously published reviews?

Furthermore, the paper requires a thorough language revision. Some corrections are suggested in the Minor comments, though the list is far from being exhaustive.

Below are my punctual comments:

(1)    Introduction: when discussing the treatment options for HNSSC, a separate paragraph should be added about current therapeutic options for HPV-positive HNC, particularly since there are arguments / conflicting results about the efficacy of cetuximab in this patient group. See for instance: https://pubmed.ncbi.nlm.nih.gov/32864799/

https://pubmed.ncbi.nlm.nih.gov/27221393/

(2)    Methods: The following sentence is not clear “It should also be noted that all information is obtained from free full texts…” Does this mean than only freely available papers were evaluated for this study? What about the articles published in traditional (subscription) journals? This might justify the low number of articles discussed in the review (21) according to your Prisma diagram.

(3)    Why only articles published after 2016 were selected? What prompted this cut-off time? Why not the past 10 years?

(4)    Refrain from using short form of verbs in a scientific paper: replace “it’s “with “it is”, “don’t” with “do not”, “isn’t” with “is not”, etc, throughout the paper and abstract.

(5)    The manuscript should undergo an English language revision, preferably by a native speaker.

(6)    When citing the author of a study in the text, use the surname only (i.e. not the initials): instead of C. Aggarwal et al, use Aggarwal et al, etc.

(7)    Please highlight the originality of your manuscript as compared to other reviews on this topic.

Minor comments:

Abstract: first sentence - Head and neck cancer (HNC) – move the acronym after ‘cancer’

Line 68 – ganglia drainage

Line 69 – this being more indicated…

Line 299 – understand the role …

Line 304 – the keywords used for the research were “head …

Line 306 – Table 1 presents the research strategies….

Line 309 – all selected articles were ….

Line 313 - …systematic reviews were selected.

Line 340 – The median rates of…

Line 360 – study, that evaluated the activity…

Author Response

“Immunotherapy in Head and Neck Cancer When, How and Why?”

Authors: Daniela Pereira, Diana Martins and Fernando Mendes

First, we would like to thank both reviewers for their coloring to allow this article to be published in the best possible way.

We appreciate all comments and suggestions for improvement.

Below are all the details of the reviews with the answers to each of the suggestions made.

Reviewers Report 2

We would like to thank you for the improvement comments, suggestions and corrections, that increased significantly the manuscript quality.

This review stands out from other published reviews on the same subject, as it encompasses the most recent therapeutic approaches aimed at patients with head and neck cancer (HNC). Knowing that the field of oncology and research around therapeutics remains in constant evolution, it becomes relevant to make known the current therapies that can, without a doubt, have proved to be promising treatment strategies for these patients.

Furthermore, this review differs from all others in that it culminates in several therapeutic strategies, rather than focusing only on one therapy, such as immune checkpoint inhibitors in isolation. The fact that this review culminates the different therapies will allow the reader to obtain a greater understanding of immunotherapy in HNC. Another interesting aspect of this review is the fact that it mentions the possible advantages and disadvantages that commonly administered therapies, such as radiotherapy and chemotherapy, may have when combining with immunotherapy.

In addition, contrary to other reviews, this review also addressed the different subtypes of HNC (HPV-associated HNC and HPV-negative HNC), to clarify how immunotherapy can act in these patients based on the individual characteristics of each tumour.

In short, this review focused on addressing all the therapeutic strategies available so far for patients with head and neck cancer in advanced stages, to clarify when, how and why they should be used, also considering PD-L1 expressions and HPV status that can undoubtedly affect the response to treatment.

  • Introduction: when discussing the treatment options for HNSSC, a separate paragraph should be added about current therapeutic options for HPV-positive HNC, particularly since there are arguments / conflicting results about the efficacy of cetuximab in this patient group:

We would like to thank you in advance for the suggestion. On page 7 (lines 299-310) the current therapies in patients with HPV-positive HNC were then described, although these are the same for patients with HPV-negative HNC.

(2) Methods: The following sentence is not clear “It should also be noted that all information is obtained from free full texts…” Does this mean than only freely available papers were evaluated for this study? What about the articles published in traditional (subscription) journals? This might justify the low number of articles discussed in the review (21) according to your Prisma diagram:

When searching the literature, the database used was only PubMed, and only full articles available for free  download or in journals subscribed by Coimbra Health School and Faculty of Medicine of the University of Coimbra would be eligible. Based on inclusion criteria’s, the eligible literature became smaller.

(3) Why only articles published after 2016 were selected? What prompted this cut-off time? Why not the past 10 years?:

Recognizing that science and research around oncology and therapeutics remain constantly evolving, it has become relevant to carry out more robust research on the most current therapies to understand how patients can currently benefit from them. It is known, therefore, that new discoveries can easily condition the treatment of cancer, making the treatment of cancer, until that moment, easily change. For this reason, articles published from 2016 onwards were selected. Another point in favor of this 6 years’ time is related to have cutting edge information only.

(4) Refrain from using short form of verbs in a scientific paper: replace “it’s “with “it is”, “don’t” with “do not”, “isn’t” with “is not”, etc, throughout the paper and abstract.: Thanks for the suggestion. Corrections were made throughout all manuscript.

(5) The manuscript should undergo an English language revision, preferably by a native speaker.: Thanks for the suggestion. We took this point into account to improve the article, and a English native read the manuscript.

(6) When citing the author of a study in the text, use the surname only (i.e. not the initials): instead of C. Aggarwal et al, use Aggarwal et al, etc.: Thanks for the suggestions. Corrections were made throughout all manuscript.

(7) Please highlight the originality of your manuscript as compared to other reviews on this topic.: This review was written with the main objective of describing what is the status of immunotherapy in patients with advanced, recurrent, or metastatic head and neck cancer.

Many of the published reviews approached the administration of immunotherapy in an isolated and rather superficial way, namely immunological checkpoint inhibitors or oncolytic vaccines or viruses. Contrary to these reviews, the present review combined different therapeutics approaches so that the reader could further improve their knowledge about head and neck cancer in a clear and faster way.

In addition, this review also addresses several aspects, such as the tumor microenvironment and evasion mechanisms, reinforced by two figures, which allows providing even more information on this topic.

On the other hand, this review culminated the different current immunotherapeutic approaches available, which allows the reader to be updated on current immunotherapy in these patients. In addition, this review has investigated more intrinsic factors exhaustively that may influence the response to immunotherapy, such as HPV status and PD-L1 expression.

Finally, the possible advantages that radiotherapy and chemotherapy may have when combined with immunotherapy were also explored.

Therefore, all these findings allowed inducing a deep reflection about the different therapeutic strategies in these patients and clarifying when, how and why immunotherapy should be used.

Based on this, this review gained advantage over all the others published so far, since it was able to address different distinct aspects and culminate them in a clear way to promote a more comprehensive and in-depth knowledge about head and neck cancer.

Abstract: first sentence - Head and neck cancer (HNC) – move the acronym after ‘cancer’

Line 68 – ganglia drainage - corrected

Line 69 – this being more indicated… - Corrected

Line 299 – understand the role …- Corrected

Line 304 – the keywords used for the research were “head … Corrected

Line 306 – Table 1 presents the research strategies…. Corrected

Line 309 – all selected articles were …. Corrected

Line 313 - …systematic reviews were selected. Corrected

Line 340 – The median rates of… Corrected

Line 360 – study, that evaluated the activity… Corrected

I would also like to thank the suggested minor comments, already corrected, to improve this article.

Round 2

Reviewer 2 Report

While the authors have addressed most comments raised by this reviewer, the language / style of the paper requires further improvements. I do not understand why the authors started replacing adequate words with synonyms that are less fitting. This is not what I meant by English revision!

Author Response

Dear Reviewer

We thank you for your email and we are absolutely delighted that the Reviewers have found our manuscript meritorious. We are grateful to the Reviewers and their constructive criticism, which has helped us to improve the manuscript.

Please find below the reply to the Reviewers’ comments.

Reviewer  Comments:

While the authors have addressed most comments raised by this reviewer, the language/style of the paper requires further improvements. I do not understand why the authors started replacing adequate words with synonyms that are less fitting. This is not what I meant by English revision.

Authors: We thank the Reviewer for this comment. We completely agree with the reviewer. A native English specialist has reviewed the manuscript. We hope it is now properly corrected for grammar, syntax and punctuation.

We hope that we have satisfactorily addressed the Reviewers’ comments.

All authors have read and approved the final version of this manuscript.

I look forward to hearing from you

Sincerely yours,

Fernando Mendes, PhD
